# Validation of the Particle-Based Multi-Analyte Technology for Detection of Anti-PhosphatidylSerine/Prothrombin Antibodies

**DOI:** 10.3390/biomedicines8120622

**Published:** 2020-12-17

**Authors:** Massimo Radin, Irene Cecchi, Silvia Grazietta Foddai, Elena Rubini, Alice Barinotti, Carlos Ramirez, Andrea Seaman, Dario Roccatello, Michael Mahler, Savino Sciascia

**Affiliations:** 1Center of Research of Immunopathology and Rare Diseases-Coordinating Center of Piemonte and Valle d’Aosta Network for Rare Diseases, S. Giovanni Bosco Hospital, Department of Clinical and Biological Sciences, University of Turin, 10154 Turin, Italy; massimo.radin@unito.it (M.R.); irene.cecchi@unito.it (I.C.); silviagrazietta.foddai@untio.it (S.G.F.); elena.rubini@unito.it (E.R.); alice.barinotti@unito.it (A.B.); linotipico@gmail.com (D.R.); 2School of Specialization of Clinical Pathology, Department of Clinical and Biological Sciences, University of Turin, 10125 Turin, Italy; 3Inova Diagnostics, San Diego, CA 92131, USA; cramirex@inovadx.com (C.R.); aseaman@inovadx.com (A.S.); mmahler@inovadx.com (M.M.); 4Nephrology and Dialysis, Department of Clinical and Biological Sciences, S. Giovanni Bosco Hospital, University of Turin, 10154 Turin, Italy

**Keywords:** antiphospholipid antibodies, antiphospholipid syndrome, anti-phosphatidylserine/prothrombin antibodies

## Abstract

Among “extra-criteria” antiphospholipid (aPL) antibodies, anti-phosphatidylserine/prothrombin (aPS/PT) antibodies, are considered a part of risk assessment strategies when investigating patients suspected of having antiphospholipid syndrome (APS). aPL detection is currently performed by solid-phase assays to identify anti-cardiolipin (aCL), anti-β2glycoprotein I (aβ2GPI) and aPS/PT antibodies, but new techniques are emerging. Among these, particle-based multi-analyte technology (PMAT), which allows the full automation and simultaneous digital detection of autoantibodies and proteins, including IgG, IgA and IgM isotypes of aCL, aβ2GPI and aPS/PT. The aim of this study was to investigate the agreement of aPS/PT testing between enzyme-linked immunosorbent assay (ELISA) and the PMAT platform. A total of 94 patients were enrolled in the study, including 71 patients with confirmed APS and 23 “aPL carriers”. aPS/PT IgG showed a moderate binomial agreement between ELISA and PMAT (k = 0.57, 95% CI 0.45–0.75), and aPS/PT IgM showed a moderate agreement (k = 0.60, 95% CI 0.45–0.75). Moreover, when considering the continuous agreement, both aPS/PT IgG and IgM showed a statistically significant correlation between ELISA and PMAT (Spearman’s correlation = 0.69, *p* < 0.001 and 0.72, *p* < 0.001, respectively). This study demonstrates that PMAT technology is a reliable method for aPS/PT IgG and IgM testing when compared to the available commercial ELISA kit.

## 1. Introduction

The antiphospholipid syndrome (APS) is an autoimmune disorder characterized by vascular thrombosis (both venous and arterial) and/or recurrent pregnancy morbidity (including miscarriages, foetal deaths, premature births and preeclampsia), associated with a persistent positivity for antiphospholipid antibodies (aPL). The current classification criteria for APS include three laboratory tests: lupus anticoagulant (LA), anti-cardiolipin (aCL) and anti-β2-glycoprotein-I (aβ2GPI) antibodies [1].

As discussed in a recent work by Negrini et al., most of the autoantibodies found in patients’ serum are directed against the plasma apolipoproteins that bind the phospholipids, especially β2GPI and prothrombin, but also other ones such as thrombomodulin, anti-thrombin, protein C, protein S, kininogens and annexin I, II and V [2]. Since the first description of the syndrome, the number of antibodies that have been associated to APS has been constantly increasing [3]. While the tests currently included in the classification criteria are able to correctly detect the great majority of the cases [4,5,6], some patients at high clinical suspicion of APS may be not identified. This is because these classification criteria exclude clinical manifestations and aPL less frequently found in patients, the so-called “extra-criteria manifestations” and “extra-criteria aPL”. aPL that are associated with APS, but are not currently included in the classification criteria, include: IgA aCL, IgA aβ2GPI, anti-phosphatidylserine, anti-prothrombin, anti-phosphatidylserine/prothrombin (aPS/PT) complex, anti-phosphatidylethanolamine, anti-vimentin, anti-phosphatidylglycerol and anti-phosphatidylinositol [2]. Among these, the aPS/PT antibodies have been proposed as an additional tool to be considered when investigating a patient suspected of having APS, particularly in the absence of routine aPL [7,8], as part of risk assessment strategies [9], or as an additional diagnostic tool when LA is inconclusive or not reliable [10]. The presence of aPL is currently assessed by solid-phase assays to identify aCL, aβ2GPI and aPS/PT antibodies. Despite significant efforts and some progresses, significant inter-laboratory and inter-assay variability still persist and considerable attempts are dedicated towards the harmonization of aPL testing [11,12]. 

New assays for detecting aPL are emerging, using a variety of approaches; some use traditional enzyme-linked immunosorbent assay (ELISA) techniques, whereas others use different platforms, which could potentially affect their diagnostic accuracy (e.g., chemiluminescence assay, thin-layer chromatography, multiline dot assay). Recently, a particle-based multi-analyte technology (PMAT) system has been developed (Aptiva™, Inova Diagnostics San Diego, USA [13]). PMAT is integrated in a fully automated random access system, which allows the simultaneous digital detection of a wide spectrum of autoantibodies and proteins [13], and these analytes are created by covalently binding antigens or antibodies to paramagnetic microparticles, as previously described [14]. More specifically, the APS reagents detect antibodies of IgG, IgA and IgM isotypes to aCL, aβ2GPI and aPS/PT specificities, respectively.

Recent studies showed a successful application of this technology when studying autoimmune diseases, such as primary biliary cholangitis, idiopathic inflammatory myopathies and also APS [13,15,16]. In this latter case, attempts to validate different methods between international laboratories have been made [12], but they focused primarily on “criteria aPL”, while more efforts need to be carried out in regard to “extra-criteria aPL”. Given this, and the growing interest of aPS/PT as a potential additional tool for APS diagnosis, the aim of this study was to investigate the agreement of aPS/PT testing between ELISA and the PMAT platform.

## 2. Experimental Section

### 2.1. Patients

After chart-reviewed consecutive patients attending the San Giovanni Bosco Hospital (Turin, Italy) in the last two years, who tested persistently positive for at least one aPL (more than 2 occasions over a time of more than 12 weeks), we enrolled a total of 94 patients who met one of the following inclusion criteria:(1)fulfilled the diagnosis of primary APS (pAPS), defined as per Sydney criteria [1];(2)fulfilled the diagnosis of secondary APS (sAPS), defined as per Sydney criteria [1];(3)tested persistently positive for aPL, with no clinical manifestations of APS (“aPL carriers”). The study was performed in compliance with the Declaration of Helsinki. Patients gave informed consent and approval was obtained from the Ethical Committee (protocol n° 0092056). Clinical and laboratory characteristics were retrospectively collected.

### 2.2. Criteria Autoantibody Detection

Complete aPL profile at inclusion in the present study included: LA, aCL IgG/IgM and aβ2GPI IgG/IgM.

All venous blood samples were collected using a 21-gauge butterfly needle, with minimal venous stasis, into Vacutainer^®^ tubes (BD, Plymouth, UK).

Serum samples, after double centrifugation at ambient temperature (2000 g for 15 min), were tested for aCL, aβ2GPI (IgG and IgM) antibodies using a commercial ELISA (QUANTA Lite, Inova Diagnostics, Inc., San Diego, CA, USA). Corresponding plasma samples were tested for the presence of LA according to the recommended criteria from the International Society on Thrombosis and Haemostasis (ISTH) Subcommittee on Lupus Anticoagulant/Phospholipid-Dependent Antibodies [17,18].

### 2.3. aPS/PT Detection

Two serum aliquots for patient were stored at −80 °C and then tested for aPS/PT antibodies (IgG and IgM) with QUANTA Lite ELISA (QUANTA Lite, Inova Diagnostics, Inc., San Diego, CA, USA) for aPS/PT antibodies (IgG and IgM) and with fully automated digital system using PMAT (Aptiva™ APS Inova Diagnostic Inc., San Diego, CA, USA) [13].

Briefly, the analytes in the assays are created by covalently binding antigens to paramagnetic microparticles. Each analyte is associated with a discrete population of particles with a unique signature that allows for their classification by an optical module. The optical module is composed of two light-emitting diodes (LEDs) units set to different wavelengths and one charge-coupled device sensor. One diode is used to classify the particles into discrete sets (populations) that are assigned to a specific analyte. This is achieved by shining light at a specific electromagnetic wavelength, while the second diode shines light at a different wavelength, selected specifically, to excite the fluorochromes present in the phychoerythrin conjugated to anti-human IgG, IgM or IgA detection antibodies. Reliability and reproducibility experiments have been performed, as previously described [13].

For PMAT and ELISA testing, we used cut-off values as per manufacturer recommendations (aPS/PT IgG/IgM ≥ 100 MFI and aPS/PT IgG/IgM ≥ 30 U, respectively).

### 2.4. Statistical Analysis

Categorical variables are presented as number (%) and continuous variables are presented as mean (S.D.). Categorical agreement was analyzed with Cohen’s kappa and degree of correlation was analyzed with the Spearman correlation. A two-sided *p*-value < 0.05 was statistically significant. All statistical analyses were performed using SPSS version 26.0 (IBM, Armonk, NY, USA). The UpSet plot was generated using the UpSetPlot library (version 0.3.0.post3) with Python (version 3.7.1). Receiver operating characteristic (ROC) curves using the qualitative results obtained by ELISA as truth and the numerical values obtained by PMAT were used to further dissect concordance between methods.

## 3. Results

A total of 94 patients were included in the study. In detail, 38 pAPS, 33 sAPS and 23 aPL carriers were recruited (mean age at study inclusion 50.2 years (±13.7), 49.1 years (±12.1) and 48.8 years (±12.8), respectively). Table 1 resumes the main demographic, clinical and laboratory characteristics of the cohort.

No differences were observed between groups when considering age and sex prevalence. The majority of patients with APS had a previous history of thrombotic manifestations (62 patients out of 71; 87%), with 37 cases of arterial thrombosis (52%) and 32 of venous thrombotic events (45%) at previous medical history. Previous pregnancy morbidity when considering Sidney criteria [1] was observed in 12 out of 42 patients (29%).

When considering previous aPL profile, 43 APS patients (61%) were triple positive for criteria aPL [1], while 13 (57%) aPL asymptomatic patients were triple positive. The most frequent aPL in the cohort was LA, positive in 81 patients (86%) out of 94 in total (with rate of positive patients as high as 91% in the aPL asymptomatic group), followed by aβ2GPI IgG/IgM (65 patients positive; 69%) and aCL IgG/IgM (63 patients positive; 67%).

Traditional cardiovascular risk factors were more present in the APS patients’ group when compared to the aPL carriers. More in detail, arterial hypertension was present in 29 APS patients (41%) vs. five (22%) aPL carriers (*p* = ns), hyperlipidemia in 25 APS patients (35%) vs. two (9%) aPL carriers (*p* = 0.014), smoking habit in 11 APS patients (16%) vs. two (9%) aPL carriers (*p* = ns) and diabetes mellitus in six APS patients (8%) vs. one (4%) aPL carriers (*p* = ns).

All patients were tested for aPS/PT (IgG and IgM) both with ELISA and PMAT.

When looking at aPS/PT testing, 63 (68%) and 50 (54%) of all patients were found to be positive for aPS/PT IgG and/or IgM with ELISA and PMAT testing, respectively (cut-off values: aPS/PT IgG/IgM ≥ 100 MFI according to PMAT and aPS/PT IgG/IgM ≥ 30 U according to ELISA). In more detail, 32 (34%) patients tested positive for aPS/PT IgG with both ELISA and PMAT testing. When considering IgM isotype, 59 patients (63%) tested positive for aPS/PT IgM when tested with the ELISA technique, while 44 patients (47%) tested positive when using PMAT. Importantly, no statistical differences were observed between groups according to diagnosis (APS, pAPS, sAPS and aPL carriers) both in terms of the rate of positive patients for aPS/PT IgG/IgM and in terms of the agreement between laboratory techniques.

Figure 1 shows the number of aPS/PT positive patients among different isotypes (IgG/IgM, IgG and IgM) comparing ELISA and PMAT techniques. aPS/PT IgG showed a moderate binomial agreement between ELISA and PMAT (k = 0.57, 95% confidence interval, CI 0.45–0.75), and aPS/PT IgM showed a moderate agreement (k = 0.60, 95% CI 0.45–0.75). When considering binomial variables, Table 2 shows in detail the numerical agreement for aPS/PT testing IgG/IgM.

When considering the continuous agreement of the analytes, both aPS/PT IgG (rho = 0.69, *p* < 0.001) and IgM (0.72, *p* < 0.001) showed a statistically significant correlation between ELISA and PMAT using Spearman’s correlation.

Figure 2 shows the continuous agreement of both IgG and IgM isotypes of aPS/PT antibodies analyzed by ELISA and PMAT. To visualize the overlap of occurrence of aPS/PT by ELISA and PMAT an upset plot was generated. Of all samples tested, 24.5% were negative and 18.1% were positive by all tests. Various combinations of positivity were observed (see Figure 3). Lastly, in order to understand the differences between the methods, ROC curves were created comparing ELISA as the reference method with PMAT. Results obtained from ELISA were used as a binary classifier, and used to generate ROC curves using the numerical results obtained by PMAT (see Figure 4). Both ROC curves were similar, with area under the curve values of 0.883 (95% CI 0.817–0.948) for IgG and 0.818 (95% CI 0.740–0.896) for IgM.

## 4. Discussion

The detection and quantification of disease-associated biomarkers and autoantibodies are central steps in the management of many autoimmune diseases, with several autoantibodies’ tests being a fundamental part of this process. In fact, diagnosis, risk assessment, follow-up and, in some cases, treatment are deeply influenced by immunological testing. Consequently, the reliability of autoantibodies measurement is crucial for the overall optimum patient care and management [19]. ELISA-based assays have been historically used for the identification of aPL, often in specialized immunological laboratory. However, autoantibody testing is now commonplace, with an increasing tendency towards more automated methods in larger number of laboratories. On the other hand, in parallel with the increasing number of immunological tests performed, we are witnessing an increase in the number of available techniques [19]. With new techniques emerging, the main concern of the treating clinicians when ordering aPL testing is the reliability and comparability of the obtained results in different laboratories using different techniques [20].

Although laboratories are required to participate in external quality assurance to ensure the comparability of the results, the quantification of aPL remains challenging, due to the nature of aPL testing. Immunoassays rely on detecting the binding of aPL to its natural antigen. As antigens associated with autoimmune diseases are typically large proteins with multiple epitopes, different immunological testing can detect different epitopes of the same autoantibody. Consequently, autoantibodies with different epitope specificities will often produce discrepant results, depending on the method of selectivity.

The topic of the comparability of solid-phase aPL testing between real world and core laboratories from the Antiphospholipid Syndrome Alliance for Clinical Trials and International Networking (APS-ACTION) has been recently addressed in an international clinical database and repository analysis. When analyzing 497 registry samples which underwent confirmatory aPL tests, they found an overall categorical agreement between the inclusion and core lab values, as expressed by Cohen’s kappa coefficients, ranged between 0.61 and 0.80 (as substantial agreement). The correlation between quantitative results in the aCL and aβ2GPI was better for IgM and IgA compared to IgG (Spearman rho 0.789 and 0.666 vs. 0.600 for aCL and rho 0.892 and 0.744 vs. 0.432 for aβ2GPI) [12]. These findings showed that, despite the enormous efforts in the harmonization of aPL testing, yet some degree of heterogeneity exists, even when comparing results coming from the same method (ELISA).

Our data aim to contribute to the ongoing debate investigating the comparability of aPL testing data when comparing the transability of results moving forward to ELISA technique to an innovative fully automated PMAT system.

In fact, not only may the numerical values of different methods show marked variations, but techniques that aim to detect the same analyte should also produce results with a consistent relationship [21]. The lack of this relationship in values, and thus a poor correlation between methods, represent a remarkable impediment for standardization and results interpretation [19]. Compared to the classification criteria markers, for which a wealth of data is available on how different assays compare, very little is known about the concordance of aPS/PT antibodies among methods. This is largely contributed to the limited availability of aPS/PT assays [22,23].

In this study, we found that level of agreement for aPS/PT testing ranges from good to excellent when comparing the traditional calcium-dependent ELISA technique with the innovative fully automated PMAT system.

In 2013, our group participated to the validation of a commercially available kit to detect aPS/PT in a cohort of systemic lupus erythematosus patients [24]. In this study, we further contributed to the topic of aPS/PT testing harmonization across different laboratory techniques, comparing the performance of ELISA with an emerging more automatized method, such as PMAT.

Our results are in line with a recent study by Elbagir and Colleagues [25], in which the authors assessed the performance of the PMAT system compared to EliA system, based on fluorescence enzyme immunoassay for the identification of different aPL isotypes in a large cohort of Sudanese and Swedish women [25]. While the need for routine testing for aPS/PT is still under debate, there is overwhelming clinical evidence that these antibodies are associated with thrombosis and pregnancy loss in patients with APS and that their presence increases the risk of developing such an event [21,26,27]. Data coming from two available systematic reviews [23,28] involving about 10,000 subjects have shown a strong association between aPS/PT and the clinical manifestations of APS. With the available level of evidence, aPS/PT testing can be considered a robust test applicable in the management of patients suspected of APS, which is also beyond the scope of this research. Therefore, the technical assessment of available methods for the detection is timely.

Finally, given the robustness of PMAT, one question still remains open. Why do we need more automatized methods for the management of autoimmune diseases such as APS?

The recognition of the molecular diagnosis to guide tailored treatment strategies has radically changed the prognoses of patient with cancer, improving patient outcomes and quality of life. These improvements have been paralleled by the economic benefits to the health care provider. Critical to precision strategies has been the identification of different of sub-groups of patients sharing peculiar serological or histological characteristics. For decades, across the range of autoimmune diseases, there has traditionally been a one-size-fits-all strategy for patient management.

The availability of more automatized method with high-throughput technologies, potentially linked with machine learning strategies, represents a necessary critical approach that will use ‘big’ data to stratify subjects with autoimmune diseases, and move towards a personalized management, which have been shown to be undoubtedly effective in cancer.

One significant limitation of our study is that we did not include a control population that would allow for assessing the specificity. Secondly, the cross-sectional designed of the study and its sample size might represent further limitations to be addressed in future researches. One should also acknowledge that, while the level of agreement ranges from good to excellent, some differences between IgG and IgM exist. However, the higher rate of positivity for IgM isotype when compared to PMAT is not totally unexpected, as the high sensitivity of ELISA-based test for IgM isotype has been largely debated, also outside the field of APS [29,30]. Moreover, some works highlighted how the role of the IgM class of aPL is still debated, since it seems to be less often associated with APS clinical manifestations than the IgG class [31,32,33]. From this perspective, if the results will be validated in larger cohort, implementing the use of the PMAT system might lead to better diagnostic accuracy, overcoming some technical limits intrinsically linked to ELISA.

## 5. Conclusions

While additional studies based on larger cohorts are needed to fully assess the assay performance on the novel PMAT system for the measurement of autoantibodies in APS, the results of this study demonstrated that the PMAT technology represents a reliable laboratory technique with great potential in APS setting and for aPS/PT detection.

In addition, when compared to ELISA, the approach PMAT offers for simultaneous and potentially time-effective methods for detecting a spectrum of antibodies in patients suspected of APS represents a leading way to improved biomarker disease profiling, and to improving our management of patients suspected of having the syndrome.

## Figures and Tables

**Figure 1 biomedicines-08-00622-f001:**
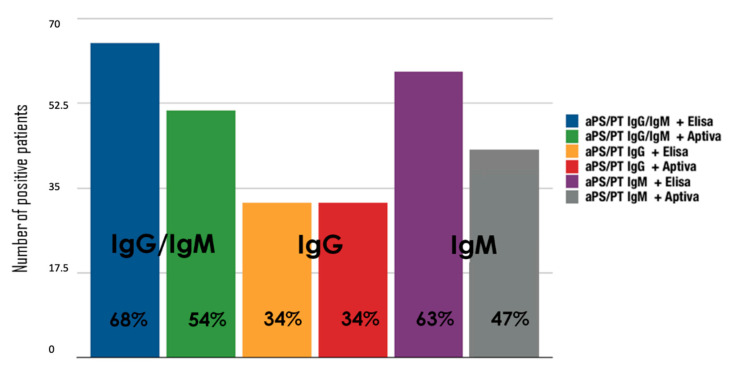
Number of positive patients with enzyme-linked immunosorbent assay (ELISA) and particle-based multi-analyte technology (PMAT) testing. aPS/PT means antiphosphatidylserine/prothrombin antibodies; ELISA, enzyme-linked immunosorbent assay.

**Figure 2 biomedicines-08-00622-f002:**
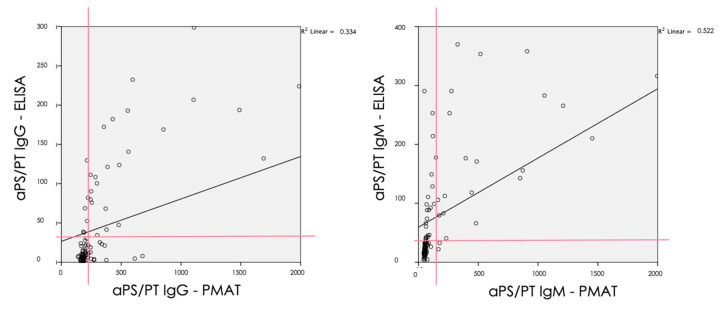
Correlation analysis with the Spearman Model of aPS/PT (IgG and IgM) antibodies between ELISA and PMAT. aPS/PT means antiphosphatidylserine/prothrombin antibodies; ELISA, enzyme-linked immunosorbent assay.

**Figure 3 biomedicines-08-00622-f003:**
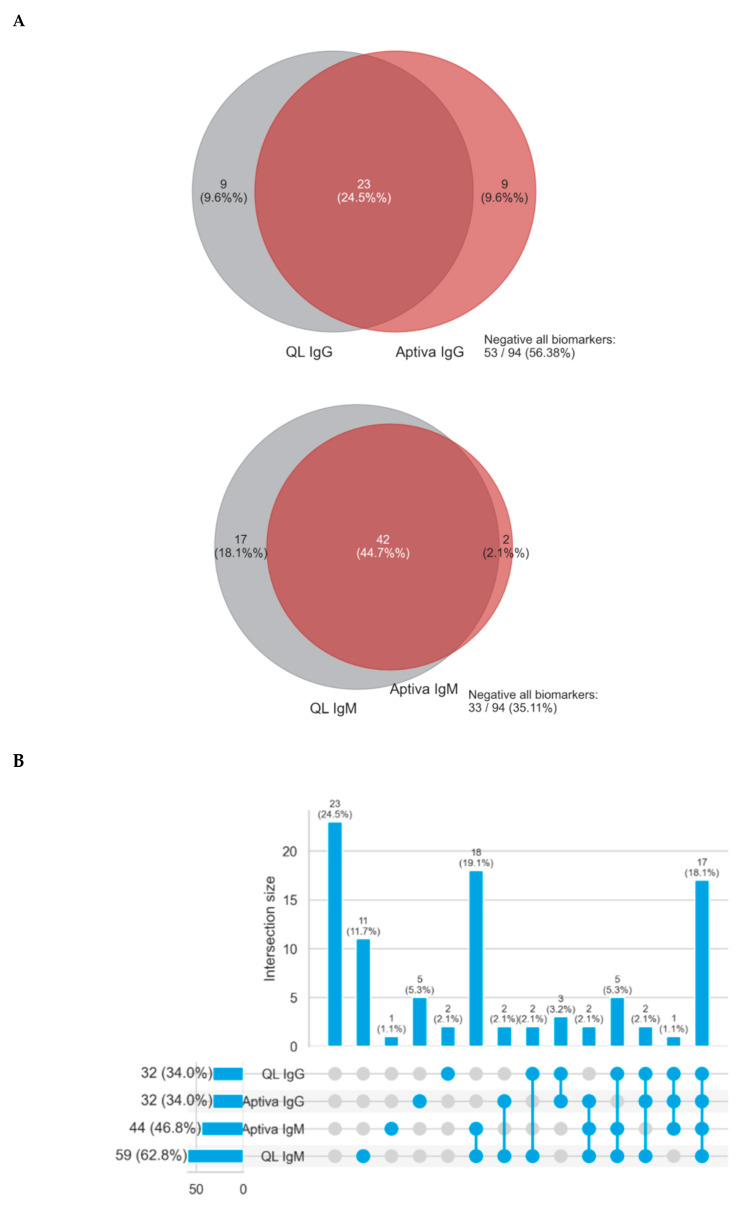
Agreement between ELISA and PMAT for the detection of anti-PS/PT antibodies. (**A**) shows Venn diagrams for IgG and IgM. In (**B**) An Upset plot to visualize the overlap of positivity for aPS/PT antibodies using ELISA and PMAT. aPS/PT means antiphosphatidylserine/prothrombin antibodies; ELISA, enzyme-linked immunosorbent assay.

**Figure 4 biomedicines-08-00622-f004:**
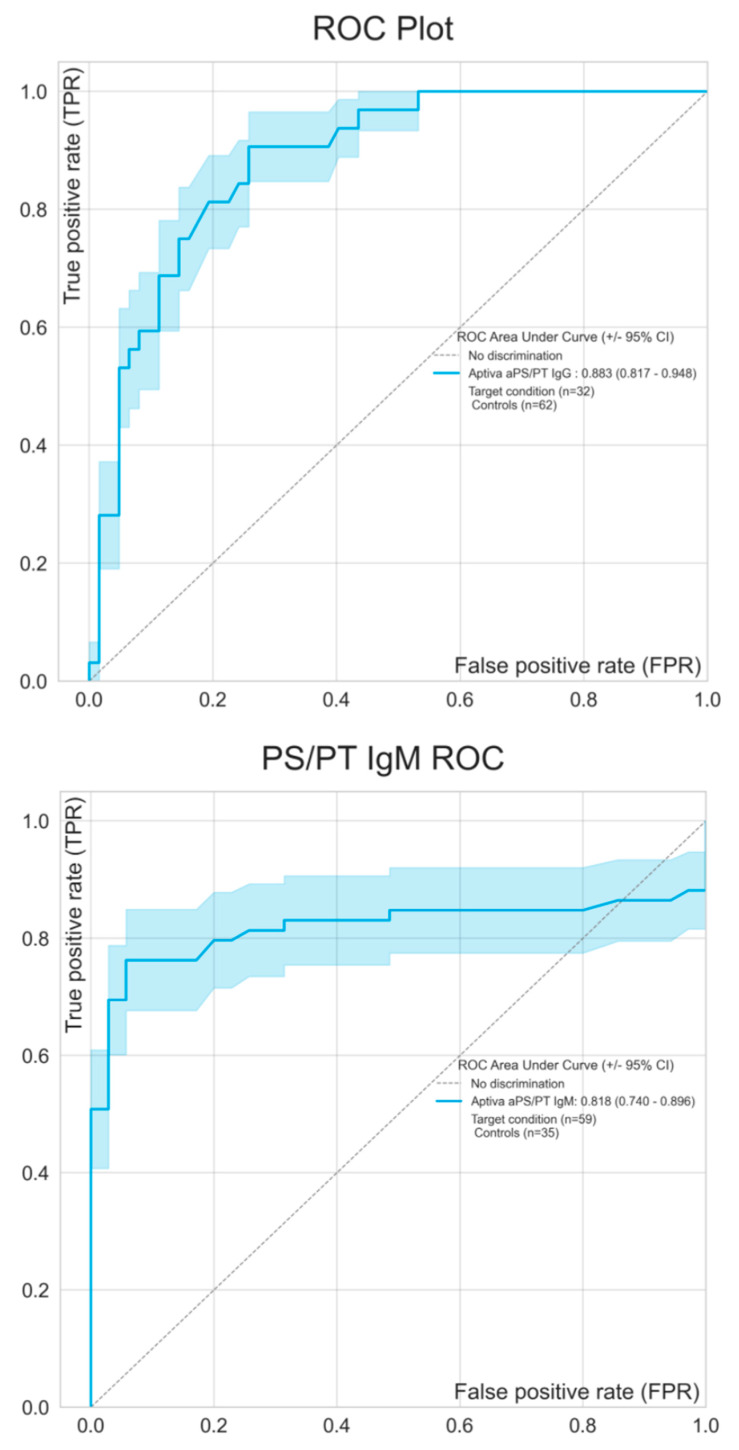
Receiver operating characteristic (ROC) analysis comparing ELISA as the reference method with PMAT. Results obtained from ELISA were used as binary classifier and employed to generate ROC curves using the numerical results obtained by PMAT. 95% confidence intervals are shown using the shaded areas. aPS/PT means antiphosphatidylserine/prothrombin antibodies; ELISA, enzyme-linked immunosorbent assay.

**Table 1 biomedicines-08-00622-t001:** Main demographic, clinical and laboratory characteristics of the patients included in the study.

	pAPS (n = 38)	sAPS (n = 33)	aPL+ (n = 23)
**Demographics and Diagnosis**			
Age at study inclusion; years (±SD)	50.2 (±13.7)	49.1 (±12.2)	48.8 (±12.8)
Females; n, (%)	24 (63)	18 (55)	17 (74)
Systemic Lupus Erythematosus; n, (%)	0	25 (76)	18 (78)
Other Autoimmune diseases *	0	8 (24)	5 (22)
**Clinical Characteristics**			
Thrombosis; n, (%)	31 (82)	31 (94)	0
Arterial; n, (%)	21 (55)	16 (52)	0
Venous; n, (%)	15 (39)	17 (52)	0
Pregnancy Morbidity; n, (%)	8 (21)	4 (12)	0
**Laboratory Profile**			
LA; n, (%)	32 (84)	28 (85)	21 (91)
aCL (IgG/M); n, (%)	25 (66)	23 (70)	15 (65)
aβ2GPI (IgG/M); n, (%)	26 (68)	24 (73)	15 (65)
Triple aPL (IgG/M); n, (%)	23 (61)	20 (61)	13 (57)
**Cardiovascular Risk Factors**			
Arterial hypertension; n, (%)	15 (39)	14 (45)	5 (22)
Hyperlipidaemia; n, (%)	14 (37)	11 (35)	2 (9)
Smoking habit; n, (%)	4 (11)	7 (23)	2 (9)
Diabetes mellitus; n, (%)	4 (11)	2 (6)	1 (4)

APS means antiphospholipid syndrome; PAPS, primary antiphospholipid syndrome; SAPS, secondary antiphospholipid syndrome; aPL+, antiphospholipid positive (“aPL carriers”); LA, lupus anticoagulant; aCL, anti-cardiolipin antibodies; aβ2GPI, anti-β2-glycoprotein-I antibodies; aPL, antiphospholipid antibodies. * Other Autoimmune diseases included: Undifferentiated Connective Tissue Disease, Mixed Connective Tissue disease and Rheumatoid Arthritis

**Table 2 biomedicines-08-00622-t002:** 2 × 2 table for aPS/PT testing using ELISA and PMAT Testing.

**Method Comparison**		**Aptiva PS/PT IgM**			**Percent Agreement** **(95% Confidence)**
		**Negative**	**Positive**	**Total**	
QUANTA Lite IgM kappa 0.60 (0.45–0.75)	Negative	33	2	35	NPA = 71.2% (58.6–81.2%)
	Positive	17	42	59	PPA = 94.3% (81.4–98.4%)
	Total	50	44	94	TPA = xx% (xxx%)
**Method Comparison**		**Aptiva PS/PT IgG**			**Percent Agreement** **(95% Confidence)**
		**Negative**	**Positive**	**Total**	
QUANTA Lite IgG kappa 0.57 (0.40–0.75)	Negative	53	9	62	NPA = 71.9% (55.6–84.4%)
	Positive	9	23	32	PPA = 85.5% (74.7–92.2%)
	Total	62	32	94	TPA = xx% (xxx%)

aPS/PT means antiphosphatidylserine/prothrombin antibodies.

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
