# Peer review of "Validation of the Particle-Based Multi-Analyte Technology for Detection of Anti-PhosphatidylSerine/Prothrombin Antibodies"

_biomedicines, 2020, doi:10.3390/biomedicines8120622_

Round 1
Reviewer 1 Report
The manuscript is interesting and well written. However, I suggest to add as reference and briefly discuss in the introduction the paper by Negrini et al. concerning antiphosholipid syndrome
Author Response
REVIEWER 1
Point 1: The manuscript is interesting and well written. However, I suggest to add as reference and briefly discuss in the introduction the paper by Negrini et al. concerning antiphosholipid syndrome
Response 1: Thank you for this enriching suggestion. The manuscript has been implemented in order to include the very interesting paper from Negrini et al, which overall improved the quality of the manuscript.
To read: “As discussed in a recent work by Negrini et al. most of the autoantibodies found in patients' serum are directed against the plasma apolipoproteins that bind the phospholipids, especially β2GPI and prothrombin, but also other ones such as thrombomodulin, anti-thrombin, protein C, protein S, kininogens and annexin I, II and V. Since the first description of the syndrome, the number of antibodies that have been associated to APS has been constantly increasing. While the tests currently included in the classification criteria are able to correctly detect the great majority of the cases, some patients at high clinical suspicion of APS may be not identified. This is because these classification criteria exclude clinical manifestations and aPL less frequently found in patients, the so-called “extra-criteria manifestations” and “extra-criteria aPL”. aPL associated with APS, but not currently included in the classification criteria comprehend: IgA aCL, IgA aβ2GPI, anti-phosphatidylserine, anti-prothrombin, anti-phosphatidylserine/prothrombin complex, anti-phosphatidylethanolamine, anti-vimentin, anti-phosphatidylglycerol and anti-phosphatidylinositol.”

Reviewer 2 Report
The antiphospholipid syndrome (APS) is an autoimmune disorder characterized by vascular thrombosis and/or recurrent pregnancy morbidity. The International Society on Thrombosis and Haemostasis (ISTH) recommends to test lupus anticoagulant, anticardiolipin and anti-b2-GP1 (IgM/IgG). However, some patients at high clinical suspicion of APS are negative for the conventional antiphospholipid antibodies. The number of antibodies that have been associated to APS is constantly increasing. Their relevance is currently debated and particularly for the anti-phosphatidylserine/prothrombin antibodies (aPS/PT). Furthermore, a variety of approaches to quantify these non-conventional antibodies are described. Very few data compare these different methods.
In this paper, the authors compared two methods: ELISA and the PMAT platform for the quantification of aPS/PT antibodies. They chose to compare the ELISA method to PMAT which allows the simultaneous detection of autoantibodies. This type of study is required.
The paper is well-written and the statistical analysis is well-done. However, several precisions have to be developed:
1/ Introduction section
The authors have to explain why they first chose to compare these two methods for the detection of aPS/PT. Is there data describing the PMAT technique for the detection of conventional aPL? Why didn’t they also compare aCL and ab2-GP1 with these two methods?
2/ Experimental section
- In their study, the authors included patients tested persistently positive for aPL with no clinical manifestations of APS. Why were these patients tested for aPL if they had no clinical manifestations? Have they been diagnosed for an autoimmune disease?
- Concerning the ethical considerations, the authors should describe if inform consent was obtained from patients to conserve their serum and use it for research. Has the biobank been declared? These data must be specified.
- Did the authors perform reliability and reproducibility experiments before comparing the two methods? These data are missing.
3/ Results and discussion
- The cut-off for the two methods has to be specified (for IgM and IgG) as well as the titer (moderate or high levels of antibodies?)
- The difference between the two methods seems to be on IgM antibodies. This result has to be discussed in regard to their clinical relevance
Author Response
REVIEWER 2
The antiphospholipid syndrome (APS) is an autoimmune disorder characterized by vascular thrombosis and/or recurrent pregnancy morbidity. The International Society on Thrombosis and Haemostasis (ISTH) recommends to test lupus anticoagulant, anticardiolipin and anti-b2-GP1 (IgM/IgG). However, some patients at high clinical suspicion of APS are negative for the conventional antiphospholipid antibodies. The number of antibodies that have been associated to APS is constantly increasing. Their relevance is currently debated and particularly for the anti-phosphatidylserine/prothrombin antibodies (aPS/PT). Furthermore, a variety of approaches to quantify these non-conventional antibodies are described. Very few data compare these different methods.
In this paper, the authors compared two methods: ELISA and the PMAT platform for the quantification of aPS/PT antibodies. They chose to compare the ELISA method to PMAT which allows the simultaneous detection of autoantibodies. This type of study is required.
The paper is well-written and the statistical analysis is well-done. However, several precisions have to be developed:
Point 1 (Introduction section): The authors have to explain why they first chose to compare these two methods for the detection of aPS/PT. Is there data describing the PMAT technique for the detection of conventional aPL? Why didn’t they also compare aCL and ab2-GP1 with these two methods?
Response 1: Thank you for suggesting this point.
PMAT system for criteria aPL was recently validated and for ELISA technique the validation of criteria aPL has come through quite some challenges (also involving a multicenter panel of experts from APS ACTION), the number of studies focusing on non-criteria aPL is still missing.
We reinforced this point in the manuscript, to read:
- Introduction section: “Recent studies showed a successful application of this technology when studying autoimmune diseases such as primary biliary cholangitis, idiopathic inflammatory myopathies and also APS. In this latter case, attempts to validate different methods between international laboratories have been made, but they focused primarily on “criteria aPL”, while more efforts need to be carried out in regard to “extra-criteria aPL”. Given this and the growing interest of aPS/PT as a potential additional tool for APS diagnosis, the aim of this study was to investigate the agreement of aPS/PT testing between ELISA and the PMAT platform.”
References:
Sciascia S, Radin M, Ramirez C et al. Evaluation of novel assays for the detection of autoantibodies in antiphospholipid syndrome. Autoimmun. Rev. 2020;102641.
Sciascia S, Willis R, Pengo V, et al. The comparison of real world and core laboratory antiphospholipid antibody ELISA results from antiphospholipid syndrome alliance for clinical trials & international networking (APS ACTION) clinical database and repository analysis. Thromb. Res. 2019;175:32–36.
Villalta D, Seaman A, Tiongson M, et al. Evaluation of a novel extended automated particle-based multi-analyte assay for the detection of autoantibodies in the diagnosis of primary biliary cholangitis. Clin. Chem. Lab. Med. 2020;58(9):1499–1507.
Cavazzana I, Richards M, Bentow C, et al. Evaluation of a novel particle-based assay for detection of autoantibodies in idiopathic inflammatory myopathies. J. Immunol. Methods. 2019;474:.
Point 2 (Experimental section): In their study, the authors included patients tested persistently positive for aPL with no clinical manifestations of APS. Why were these patients tested for aPL if they had no clinical manifestations? Have they been diagnosed for an autoimmune disease?
Response 2: Thank you for this point. Most of patients that did not experience APS clinical manifestations were tested because they suffered from an autoimmune disease and were screened for the presence of aPL. The manuscript has been revised in order to include this information, now Table 1 includes other diagnosis of the patients included in the study.
Point 3 (Experimental section):Concerning the ethical considerations, the authors should describe if inform consent was obtained from patients to conserve their serum and use it for research. Has the biobank been declared? These data must be specified.
Response 3: Thank you. Informed consent and approval from the local Ethical committee was obtained. The requested information has been added to the manuscript.
Point 4 (Experimental section): Did the authors perform reliability and reproducibility experiments before comparing the two methods? These data are missing.
Response 4: Thank you for this suggestion. Reliability and reproducibility experiments have been previous performed in order to validate the PMAT system. In order to include the clarity of the manuscript the “experimental section” has been changed accordingly, to read:
“Reliability and reproducibility experiments have been performed, as previously described. ”
References:
Sciascia S, Radin M, Ramirex C et al. Evaluation of novel assays for the detection of autoantibodies in antiphospholipid syndrome. Autoimmun. Rev. 2020;102641.
Sciascia S, Willis R, Pengo V, et al. The comparison of real world and core laboratory antiphospholipid antibody ELISA results from antiphospholipid syndrome alliance for clinical trials & international networking (APS ACTION) clinical database and repository analysis. Thromb. Res. 2019;175:32–36.
Villalta D, Seaman A, Tiongson M, et al. Evaluation of a novel extended automated particle-based multi-analyte assay for the detection of autoantibodies in the diagnosis of primary biliary cholangitis. Clin. Chem. Lab. Med. 2020;58(9):1499–1507.
Point 5 (Results and discussion): The cut-off for the two methods has to be specified (for IgM and IgG) as well as the titer (moderate or high levels of antibodies?)
Response 5: Thank you for this comment, the work has been modified accordingly. We previously reported the cut-off values for the two methods in the “Experimental section”, but as you suggested, we added them also in the results section.
To read: “…When looking at aPS/PT testing, 63 (68%) and 50 (54%) of all patients were found to be positive for aPS/PT IgG and/or IgM with ELISA and PMAT testing respectively (cut-off values: aPS/PT IgG/IgM ≥ 100 MFI according to PMAT and aPS/PT IgG/IgM ≥ 30 U according to ELISA). …”
Point 6 (Results and discussion): The difference between the two methods seems to be on IgM antibodies. This result has to be discussed in regard to their clinical relevance.
Response 6: Thank you for the suggestion. The manuscript has been revised accordingly. To read in the “Discussion” section: “…Moreover, some works highlighted how the role of the IgM class of aPL is still debated, since it seems to be less often associated with APS clinical manifestations than the IgG class. …”
References:
Kelchtermans H, Pelkmans L, de Laat B, Devreese KM. IgG/IgM antiphospholipid antibodies present in the classification criteria for the antiphospholipid syndrome: a critical review of their association with thrombosis. J. Thromb. Haemost. 2016;14(8):1530–1548.
Galli M, Luciani D, Bertolini G, Barbui T. Lupus anticoagulants are stronger risk factors for thrombosis than anticardiolipin antibodies in the antiphospholipid syndrome: A systematic review of the literature. Blood. 2003;101(5):1827–1832.
Galli M, Luciani D, Bertolini G, Barbui T. Anti-β2-glycoprotein I, antiprothrombin antibodies, and the risk of thrombosis in the antiphospholipid syndrome. Blood. 2003;102(8):2717–2723.

Round 2
Reviewer 2 Report
All my comments have been adequately addressed by the authors.